



# Opinion: Insights into updating Ambient Air Quality Directive (2008/50EC)

Joel Kuula[1], Hilkka Timonen[1], Jarkko V. Niemi[2], Hanna E. Manninen[2], Topi Rönkkö[3], Tareq Hussein[4], Pak Lun Fung[4], Sasu Tarkoma[5], Mikko Laakso[6], Erkka Saukko[7], Aino Ovaska[4], Markku Kulmala[4], Ari Karppinen[1], Lasse Johansson[1], Tuukka Petäjä[4]

[1]Atmospheric Composition Research, Finnish Meteorological Institute, Helsinki, 00560, Helsinki
[2]Helsinki Region Environmental Services Authority HSY, Helsinki, 00240, Helsinki, Finland
[3]Aerosol Physics Laboratory, Physics Unit, Tampere University, Tampere, 33100, Finland
[4]Institute for Atmospheric and Earth System Research (INAR/Physics), University of Helsinki, Helsinki, 00560, Finland
[5]Department of Computer Science, University of Helsinki, Helsinki, 00560, Finland
[6]Vaisala Oyj, Vantaa, 01670, Finland
[7]Pegasor Oy, Tampere, 33100, Finland

*Correspondence to*: Joel Kuula (joel.kuula@fmi.fi)

**Abstract.** As the evidence for the adverse health effects of air pollution continues to increase, World Health Organization (WHO) recently published its latest edition of the Global Air Quality Guidelines. Although not legally binding, the guidelines aim to provide a framework in which policymakers can combat air pollution by formulating evidence-based air quality management strategies. In the light of this, European Union has stated its intent to revise the current Ambient Air Quality Directive (2008/50/EC) to resemble closer to that of the newly published WHO guidelines. This article provides an informed opinion on selected features of the air quality directive that we believe would benefit from a reassessment. The selected features include discussion about 1) air quality sensors as a part of hierarchical observation network, 2) number of minimum sampling points and their siting criteria, and 3) new target air pollution parameters for future consideration.

## 1 Background

Air pollution continues to be one of the top-ranking mortality risk factors in the global burden on disease analysis, and different estimates have linked air pollution to 3.3-9 million premature deaths globally (Burnett et al., 2018; Lelieveld et al., 2015; World Health Organization, 2021). Air pollution consists of both gas- and particle-phased components, which originate from a variety of different anthropogenic and natural sources. Typical anthropogenic sources are related to combustion processes such as vehicular exhaust emissions and residential wood burning whereas natural sources include, for instance, wildfires and volcano eruptions (e.g. Aguilera et al., 2021; Rönkkö et al., 2017). From the health effects point of view, particulate matter (PM2.5 and PM10) is the most detrimental to human health, although trace gases such as NO2 and O3 also have a contribution (European Environment Agency, 2020). Besides regulated mass-based PM2.5 and PM10 parameters, typically evaluated non-regulated particle metrics also include particle number and surface area-based



concentrations, particle size distributions as well as black carbon or elemental carbon (BC/EC) (e.g. de Jesus et al., 2019; Wu & Boor, 2020). Exposure to air pollution occurs both indoors and outdoors although people spend most of their time indoors (World Health Organization, 2006).


To combat poor outdoor air quality, World Health Organization (WHO) has produced a series of guidelines to support policymakers in setting air quality management strategies. Although being neither standards nor legally binding criteria, the guidelines are based on expert evaluation of scientific evidence and are thus a valuable source of information. The earliest guidelines were published in 1987 and the subsequent revisions in 2000 and 2006. The newest edition, which was published

in 2021, represents the most up-to-date understanding of air pollution and its impacts on human lives (World Health Organization, 2021). In the light of this, the European Union has stated its intent to revise the current Ambient Air Quality Directive 2008/50/EC (AAQD, European Council, 2008) to resemble closer to that of the new WHO guidelines. The purpose of this article is to provide an expert opinion on how the AAQD should be developed taking into account the latest technological and scientific advances in air quality monitoring.

**2 Air quality sensors as a part of hierarchical observation network**

In the current AAQD documentation, air quality observations comprise of four modes (Table 1) as follows: 1) fixed measurements, 2) indicative measurements, 3) modelling, and 4) objective estimation. In the current hierarchy the uncertainties are shown in a descending order from the most stringent and accurate to the least stringent and accurate. The two most accurate modes – fixed and indicative measurements – are based on the actual measurements whereas the two least

accurate modes – modelling and objective estimation – are based on mathematical and expert estimations of the concentrations. The main weakness of the measurement-based observation modes is their expensiveness and laboriousness; establishing and maintaining a measurement station, especially a fixed measurement-classified station, requires considerable resources, leading to a scarcity of stations and poor spatial coverage of measurements. Modelling and objective estimation observation modes require far less continuous effort and can cover essentially any spatial domain but provide potentially less

reliable, mathematically derived data. The hierarchical observation framework, in which different tier-level observation modes can be used in parallel to complement each other's strengths and weaknesses, is well justified; the authority conducting monitoring may use a combination of techniques that best fits its needs and resources.

Currently, the devices used for measurement-based online observations are almost exclusively limited to fixed measurement

types. This is at least partly due to the long and costly process of acquiring a device type approval, which is mandatory if regulatory measurements are to be made. Type approval means that the measurement device adheres to the specifications set forth in standard DIN EN 15267, and that the adherence to this standard has been verified by an accredited laboratory (TÜV Rheinland). Because the type approval process is the same and the cost is the same for both fixed and indicative





measurement devices, there is little incentive for a company to pursue other than the more stringent and accurate fixed
measurement classification. This is problematic, when considering the recent emergence and proven usefulness of air quality
sensors (e.g. Petäjä et al., 2021). Although sensors have no formal definition, they are typically perceived as small,
standalone devices, which are easy to use and easy to deploy within the city infrastructure due to their wireless
communication features. Furthermore, their cost is usually a fraction (e.g. < 5000 EUR) of a conventional fixed
measurement monitor. Air quality sensors and their respective performances have been studied intensively for several years
(e.g. Alfano et al., 2020), and they have been used successfully in several voluntary, non-regulatory applications, such as in
the wildfire smoke map provided by the U.S. EPA (https://fire.airnow.gov). The consensus within the research community is
that, while sensors are unlikely to be a direct replacement to the established fixed measurement monitors, their capability to
cost-efficiently complement existing monitoring networks as indicative measurement devices is a novel and valuable feature
(Peltier, 2020). However, the current formulation and procedure for device type approval does not facilitate extensive
integration of sensors into regulatory air quality management strategies, and a new testing protocol is needed if this is to be
changed.

Although the testing protocol should be less exhaustive for the companies to apply the device type approval with a lower
threshold, it is equally important that no major compromises are being made with respect to the quality criteria of the testing
protocol; the sensors should still be able to measure target pollutants with an adequate certainty over a certain, for example
1-year timespan (Kaduwela & Wexler, 2021). Additionally, an important point of focus is the sensor intra-unit precision.
Distinguishing sensor imprecisions from the true spatial variability of pollutant concentrations is paramount as one of the
most useful applications for sensors is the ultra-dense monitoring networks (Popoola et al., 2018). In practice, if the
uncertainty of a measurement from one node to another is too large, identifying and assessing the local concentration
hotspots and emission sources will not be possible. When considering dense monitoring networks alone, it could be argued
that the absolute sensor accuracy is secondary to its intra-unit precision.

When testing and evaluating sensors, the manufacturer should make clear whether the data generated by a sensor is based on
an actual measurement of the pollutant itself or whether it is based on the combination of, for example, machine learning and
secondary data (Li et al., 2020). This is important to ensure transparency. If the sensor relies heavily on data post-processing
and advanced conversion methods, the data should be treated as modelling data. In our view, this is because the data then
resembles closer to a mathematical prediction rather than an independent measurement. It can be debated where exactly the
boundary between a measurement and a prediction is; some empirical corrections may be justifiable and reasonable
(Schneider et al., 2019). However, as multiple studies have shown that sensors may entail detrimental flaws (Castell et al.,
2017; Giordano et al., 2021), it is very important to distinguish between a real and valid observation and, for example, an AI-
based extrapolation of the originally insufficient and inaccurate data.



The authors are aware of the technical specification CEN/TC 264/WG 42 - Ambient air – Air quality sensors being under development, and publication of this specification is awaited with interest.

## 3 Determining the minimum amount of sampling points

In the European Union, the Member States define their zones and agglomerates in which air quality monitoring takes place. Typically, the division between areas follows the municipality and city boarders, although joint efforts, where neighbouring municipalities or cities conduct air quality monitoring together, are also possible. In general, the minimum number of fixed sampling points for a specific area is then determined according to the population of that area, as shown in Table 2. As population is the sole factor determining the minimum number of sampling points, this may lead to cases where the number of minimum sampling points between two vastly different zoners or agglomerates is strikingly similar. For example, the Finnish Lapland – one of the monitoring zones in Finland – covers an area of approximately 100 000 km2 and has 175 000 inhabitants, whereas the Helsinki Metropolitan area – also a monitoring zone in Finland – covers an area of 1 500 km2 and has 1.2 million inhabitants. According to Table 2, if the maximum concentrations are between the upper and lower assessment thresholds, the Lapland monitoring zone requires a total of 2 sampling points (1 per 87 500 inhabitants) whereas the Helsinki Metropolitan area requires 5 sampling points (1 per 240 000 inhabitants). In our view, the proportions of the required sampling points are imbalanced, and the difference in required sampling points should be greater between the two areas. A more sophisticated approach would be to connect the minimum number of sampling points to the population density rather than to just population alone. Population density data is readily available on-line (https://ghsl.jrc.ec.europa.eu/ghs_pop2019.php). Implementation of this connection could be done in a multitude of ways, and it should be ensured that the scaling is appropriate; on average, the number of sampling points should increase rather than decrease. Taking population density into account the determination of the amount of observation points would track more closely the real exposure citizens are subjected to and thus the directive would better fulfil its main purpose of protecting human health.

## 4 Siting criteria of sampling points

In the current AAQD, the definitions and instructions regarding the siting of sampling points are in some parts poorly formulated. This has been outlined exceptionally well by Maiheu & Janssen (2019) in their report *Assessing the spatial representativeness of AQ sampling points*. Some examples of the poor formulation include the use of vague terms such as "some metres away" and "edge of major junction", which both can be interpreted in a multitude of ways. We propose here that rather than trying to define the ultimate set of all-encompassing siting guidelines, it is worth considering to what extent it would be appropriate to rely on the expertise and judgement of the local air quality authorities themselves. The potential abuse of liberty with respect to artificially improving air quality by situating the stations to only low concentration areas is a



real problem, and some boundary conditions are necessary. But in reality, practicalities such as availability of power and local regulations have a significant impact on the final site location. As the technology evolves, sensors may eventually play

a key role in solving this dilemma, but as of yet, the clear and well-defined siting criteria and practical challenges related to the deployment of monitors are in odds with one another. With respect to solving the problem of "spatial representativeness" and prioritising siting criteria, measurements at two different sites instead of only one solves these problems in many cases; there is no need to make decision between two equally worthy measurement locations if the amount of sampling points allows both locations to be covered. Fundamentally, the necessity of siting criteria arises from the scarcity of measurements,

and the hierarchical monitoring approach is the right tool to address it. This further underlines the need and utility of dense sensor networks.

## 5 New target parameters for future consideration

The primary motivation of ambient air quality monitoring is the protection of human health. Epidemiological studies have shown that particulate matter is the dominant pollutant causing morbidity and premature deaths (European Environment

Agency, 2020); however, it is still unclear to some degree, what the explicit mechanisms driving the adverse health effects are (Fiordelisi et al., 2017). Mass-based parameters PM2.5 and PM10 have been used as target parameters due to the clear evidence of their association with adverse health effects (Burnett et al., 2018). Ultrafine particles have also been considered a factor, but the current literature suggests that besides their short-term association with inflammatory and cardiovascular changes, there is insufficient evidence showing that they drive negative health effects independently of other particulate

matter constituents (Ohlwein et al., 2019). Exposure to black carbon also entails negative health effects, and it has been shown to be an important detrimental component of PM2.5 (Yang et al., 2019). Additionally, the climatic effects of black carbon further underline its importance regarding air quality monitoring (Bond et al., 2013). Perhaps an often-overlooked parameter is pollen, which causes irritation symptoms to as many people as one in four in Europe each year (Bauchau & Durham, 2004). Pollen originates from plants, and besides reducing the exposure to it by personal behavioural changes, it is

questionable what type of mitigation measures could be taken to reduce its concentrations. Nevertheless, out of all the particulate matter parameters and constituents, the adverse health effects of pollen are the most visible and evident in people's daily lives. Similarly to sensor technology, measurement techniques for pollen have advanced, and monitors capable of online concentration measurement and species identification are currently available in the markets. Lastly, the monitoring of ambient aerosol parameters that are included in European emission regulations could significantly improve

both the understanding of emission sources and the emission mitigation actions; for instance, regulatory emission measurements made for vehicles in road traffic include the measurement of exhaust solid particle number, which is currently not monitored from traffic-influenced urban air.



Monitoring additional particulate parameters would undoubtedly be useful from the scientific point of view, and the newly

published WHO guidelines supports this as it recommends monitoring of ultrafine particles (UFP) and BC/EC (World Health Organization, 2021). However, monitoring of the additional parameters would require new and extensive standardization with respect to the new parameter-specific instrumentation and expensive investments by the monitoring bodies to these new instruments. Therefore, it may be more appropriate to suggest that the scientific community monitors these additional parameters with supporting, research-type Supersite stations, which remain outside of the regulatory monitoring framework.

Naturally, air quality authorities capable of conducting more extensive, voluntary-based monitoring are encouraged to do so, and initiating the preparation for instrument standardization would be useful regardless of the direction in which the AAQD is headed in the future. As illustrated in Fig. 1, the idea of a Supersite station is to support and complement existing regulatory monitoring and to provide novel insights into air quality (e.g. Kulmala et al., 2021). The instrumentation could cover measurements for the additional parameters described previously as well as for the more advanced parameters such as

chemical composition of particles. Moreover, due to the extensive measurement arsenal, new observation technologies could be benchmarked and calibrated rapidly. The more in-depth data generated by the Supersites would enable scientists and policymakers to assess the needs and directions of future air quality regulation.

## 6 Summary and Outlook

Accurate and reliable fixed measurement monitors are, and should remain, the backbone of regulatory air quality monitoring.

Nevertheless, technological development of air quality sensors is advancing rapidly. The full benefit of the new observational capacities of small sensors can be obtained when a hierarchical observation system is considered (e.g. Hari et al., 2016). This would allow verification of the sensor network data quality against the higher quality observation sites in the vicinity. The hierarchical network would allow science-based development and rapid deployment of novel air quality devices as soon as they become available.


Instead of determining the siting criteria to match all the possible urban environments, more trust should be placed upon local air quality expert analysis, and in some cases applying multiple sensors can be the way forward. Furthermore, the population density should be considered as the basis for determining the required amount of air quality observations. This will lead to an optimized air quality monitoring network providing more representative data in terms of population exposure.

At the same time, a specific focus should be placed upon constraining the intra-sensor variability to ensure that the true spatial variability of concentrations and sensor imprecision are not mixed with each other.

Monitoring of additional air quality parameters, such as ultrafine particles and black carbon, would be beneficial in terms of better understanding air pollution, and research-type Supersite stations operated by scientific community or, where possible,

local air quality authorities could provide the platform for such monitoring. While supporting and complementing regulatory





observation network, the more in-depth data generated by the Supersites could help in navigating the future air quality directive requirements.

## Data availability

No data sets were used in this article.

## Author contributions


J.K, H.T., and T.P. drafted the preliminary version of the article. All authors contributed to the ideation, commenting, writing, and editing of the text. J.K. finalised the manuscript.

## Competing interests

Markku Kulmala and Tuukka Petäjä are members of the editorial board of Atmospheric Chemistry and Physics. The peer-

review process was guided by an independent editor, and the authors have also no other competing interests to declare.

## Financial support

This work is funded under the European Union's Horizon 2020 research and innovation programme, Grant agreement No 101036245 project RI-URBANS (Research Infrastructures Services Reinforcing Air Quality Monitoring Capacities in European Urban & Industrial AreaS),  Business Finland and participating companies via CITYZER (3021/31/2015,

2883/31/2015) and BC Footprint (528/31/2019, 530/31/2019) projects, European Regional Development Fund: Urban innovative actions initiative (HOPE; Healthy Outdoor Premises for Everyone, project no: UIA03-240), regional innovations and experimentations funds AIKO, governed by the Helsinki-Uusimaa Regional Council (project HAQT, AIKO014), Academy of Finland Flagship funding (grant no. 337549, 337551, 337552).

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

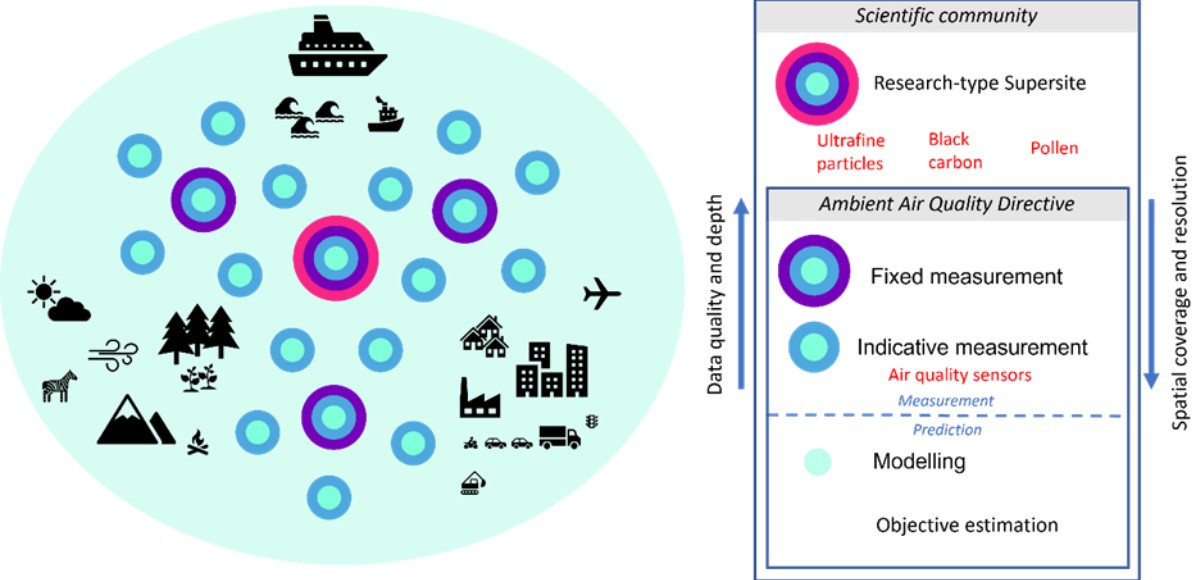

**Figure 1: Concept map of air quality monitoring, which combines both regulatory measurements as well as the research-type Supersite stations.**





**Table 1. Current AAQD observation modes and their measurement uncertainties adapted from the AAQD (European Council, 2008).**

| | SO$_2$, NO$_2$, NO$_x$ and CO | Benzene | PM$_{2.5}$, PM$_{10}$ and Pb | O$_3$ and related NO and NO$_2$ |
|---|---|---|---|---|
| **1) Fixed measurements [1]** | | | | |
| Uncertainty | 15 % | 25 % | 25 % | 15 % |
| Minimum data capture | 90 % | 90 % | 90 % | 90 % during summer 75 % during winter |
| Minimum time coverage | | | | |
| - when background and traffic | - | 35 % ([2]) | - | - |
| - industrial sites | - | 90 % | - | - |
| **2) Indicative measurements** | | | | |
| Uncertainty | 25 % | 30 % | 50 % | 30 % |
| Minimum data capture | 90 % | 90 % | 90 % | 90 % |
| Minimum time coverage | 14 % | 14 % ([3]) | 14 % ([4]) | > 10 % during summer |
| **3) Modelling uncertainty** | | | | |
| Hourly | 50 % | - | - | 50 % |
| Eight-hour averages | 50 % | - | - | 50 % |
| Daily averages | 50 % | - | not yet defined | - |
| Annual averages | 30 % | 50 % | 50 % | - |
| **4) Objective estimation** | | | | |
| Uncertainty | 75 % | 100 % | 100 % | 75 % |

[1] Member States may apply random measurements instead of continuous measurements for benzene, lead and particulate matter if they can demonstrate to the Commission that the uncertainty, including the uncertainty due to random sampling, meets the quality objective of 25 % and the time coverage is still larger than the minimum time coverage for indicative measurements. Random sampling must be evenly

distributed over the year in order to avoid skewing of results. The uncertainty due to random sampling may be determined by the procedure laid down in ISO 11222 (2002) 'Air Quality — Determination of the Uncertainty of the Time Average of Air Quality Measurements'. If random measurements are used to assess the requirements of the PM10 limit value, the 90,4 percentile (to be lower than or equal to 50 µg/m3) should be evaluated instead of the number of exceedances, which is highly influenced by data coverage.

[2] Distributed over the year to be representative of various conditions for climate and traffic.

[3] One day's measurement a week at random, evenly distributed over the year, or eight weeks evenly distributed over the year.

[4] One measurement a week at random, evenly distributed over the year, or eight weeks evenly distributed over the year.



**Table 2. Minimum number of sampling points per zone or agglomerate as a function of population (European Council, 2008).**

| Population of zone or agglomeration in thousands | If the maximum concentrations exceed upper assessment threshold | | If the maximum concentrations are between the upper and lower assessment thresholds | |
|---|---|---|---|---|
| | Pollutants except PM | PM (sum of $PM_{2.5}$ and $PM_{10}$) | Pollutants except PM | PM (sum of $PM_{2.5}$ and $PM_{10}$) |
| 0-249 | 1 | 2 | 1 | 1 |
| 250-499 | 2 | 3 | 1 | 2 |
| 500-749 | 2 | 3 | 1 | 2 |
| 750-999 | 3 | 4 | 1 | 2 |
| 1000-1499 | 4 | 6 | 2 | 3 |
| 1500-1999 | 5 | 7 | 2 | 3 |
| 2000-2749 | 6 | 8 | 3 | 4 |
| 2750-3749 | 7 | 10 | 3 | 4 |
| 3750-4749 | 8 | 11 | 3 | 6 |
| 4750-5999 | 9 | 13 | 4 | 6 |
| $\geq 6000$ | 10 | 15 | 4 | 7 |