# Peer review of "Opinion: Insights into updating Ambient Air Quality Directive (2008/50/EC)"

_Atmospheric Chemistry and Physics, 2021_

## Author Comment (AC1)

We thank the reviewers for taking time to read the article thoroughly and their valuable comments. We have answered all comments and we think the quality of the article has significantly improved in the review.

**Reviewer #1**

General Comments

I do not feel that the paper presents a sound or compelling argument for updating the Ambient Air Quality Directive (AAQD). The concepts presented do not appear to be novel, and the conclusions do not appear to be reached from a full appreciation of the context of the AAQD and the reality of what different member states do.

Specific Comments

WHO Guidelines

The paper is predicated on the Ambient Air Quality Directive (AAQD) being updated in response to the WHO guidelines. A citation for the statement in lines 17/18 and 41/42 would thus be helpful.

*A citation referring to the Inception impact assessment -report, which describes the Commission's reasoning behind AAQD revision, was added to lines 17/18 and 41/42.*

Context of the AAQD

The AAQD has undergone substantial recent consultation, and I believe that the main issues raised in the paper were flagged then. It is unclear how the paper interfaces with this process, and acknowledgement of the consultation may be helpful.

*In addition to the reviewer's referred expert consultation (survey), there was a public open consultation forum (open Sep 23 – Dec 16, 2021) where all citizens and the wider community were welcome to express their views. The preprint of this article was submitted to the open consultation forum. We think that open discussion taking into account all viewpoints is important in order to compile the best possible Air quality directive that would be useful for the upcoming years and decades.*

*Also, the recent expert survey was biased in terms of respondent's represented Member State and type (e.g. local authority, academia) and it may not be an accurate summary of the views held on AAQD revision. In particular, the proportion of responses submitted by the academia was low, ranging from 2.9 to 8.5% depending on the question. If an objective revision is desired, the opinions expressed in this manuscript are valuable with respect to improving the survey statistics.*

*A note and a link to the open consultation forum and expert survey results were added to the manuscript:*

*"There has been two open consultation rounds in which experts (open from Feb 1 to Mar 1, 2021) and wider community (open from 23 Sep to 16 Dec 2021) were welcome to express their views on the AAQD revision. The results of the expert survey can be found here: https://ec.europa.eu/environment/air/quality/documents/20210831_SR9%20Phase%201%20Report _TechAnnex.pdf."*

I believe that the paper would also benefit from greater cognisance of the wider context of the AAQD and how measurements fit within this. Observations are a tool for achieving the broader AAQD aim of improving and maintaining air quality. This is relevant because networks of sensors are already widely and routinely across Europe in the context of improving air quality. Similarly, passive instruments which provide less time resolution at lower cost are also used for this purpose. The section in lines 45 to 75 culminates with the statement that sensors are prohibited from integration into "regulatory air quality management strategies". It is unclear what this phrase means since the absence of accreditation as either a fixed or indicative method does not (as evidenced by current work across Europe) preclude the use of sensors in air quality management strategies.

*As sensors and sensor networks are already widely used across Europe, we believe it is reasonable to propose these to be harmonized through legislative means.*

*Measurement instrumentation, which does not adhere to its suitability criteria set in regulation, is non-compliant. This means that the legal obligations set by the regulation cannot be fulfilled with such instrumentation, no matter how useful they are. To make this clearer, "regulatory air quality management strategies" has been rephrased as "air quality management strategies aiming to fulfill the legal obligations set by the AAQD"*

Sensor Evaluation

CEN Working Group 24 is currently working on performance specifications for sensors. The gases Technical Specification is currently out for CEN Enquiry - the PM one is underway. These Technical Specifications are for sensor systems only, and do not address networks of sensors - it is likely that will come but at some stage in the future. I do not believe that there is anything within AAQD that precludes the use of sensor systems or networks of sensors if they meet the Data Quality Objectives of any future CEN standards. Lines 78 and 79 suggest a preference for a more streamlined approach to performance testing but does not provide explicit suggestions of how this would work. My understanding is that WG42 is already mindful of the burden on manufacturers for performance testing, but ultimately robust performance testing is essential. The paper would benefit from a clearer explanation of how the current approach might be improved.

*We have no doubt that the WG42 is already aware of the nuances related to sensor standardization and that the subject of testing laboriousness has been discussed in detail. The intent here is to reinforce the perception of the validity of this action and underline that, if sensors are to be part of regulatory air quality measurements, their testing protocol must be made such that companies are willing to pursue type-approvals for their products. It is also worth noting that more streamlined protocol would most likely accelerate the evolution of sensor markets and development of technology, which is desirable.*

*To make explicit suggestions for streamlining the testing, the following was added to the manuscript:*

*"When considering the type-approval process of PM measurement systems specifically (EN16450), which is perhaps the most laborious of the classical target pollutants (PM, $O_3$, $NO_2$, $SO_2$, and CO), a straightforward way to ease the burden of testing would be to replace the use of gravimetric reference measurement with a type-approved automated reference monitor. This would eliminate much of the manual work in field tests (e.g. filter weighing) and thus reduce cost. Another simple way to ease the burden of testing could be to reduce the minimum amount of 24h-averaged measurement samples (currently 160) required for the equivalency comparison."*

Minimum number of Sampling Points

The statement in Line 102 is incorrect. The concept here seems sensible, but the comparison between Helsinki and Lapland seems overly simplistic and some other worked examples would provide a more compelling case.

*To be more precise, line 102 has been rephrased as follows:*

*"Typically, the division between areas follows the administrative unit boarders although joint efforts, where neighboring units conduct air quality monitoring together, are also possible."*

*A similar example from Norway was added to the manuscript:*

*"A fairly similar example to that of the Finland can be found in Norway between the Oslo metropolitan agglomerate (5 sampling points; 1 per 206 000 inhabitants) and Troms and Finmark zone (2 sampling points; 1 per 117 000 inhabitants)."*

*Related data can be found at https://eeadmz1-cws-wp-air02.azurewebsites.net/index.php/users-corner/.*

Siting Criteria

This section (lines 120 to 135) seems to be predicated on the assumption, which is set explicitly at line 134, that siting criteria are necessary because of the scarcity of sites. I believe that this assumption is incorrect. There is significant deviation regarding how instruments are sited across Europe, with different areas having favoured different approaches. It is also far from uncommon for there to be legal discussions (outside of the CJEU) regarding the applicability of data representing a specific location. Increasing site number does not solve systematic differences in siting approach. Neither does placing more reliance on local judgement. Since legal issues surrounding air quality measurements are unlikely to stop soon, increasing the number of sampling points potentially places more emphasis on siting criteria rather than weakening the requirement for them.

*We agree that the problems associated with the siting process are hard to resolve, and it is unlikely that any single factor will be able to remedy the situation completely. Nevertheless, we believe that seeking for an improvement is worthwhile.*

*It is clear, as evidenced by the current reality of deviating siting approaches, that the siting guidelines would benefit of a more precise formulation; however, as we point out in the manuscript, it is probable that strictly unidimensional rules will be in odds with the practicalities related to the deployment of measurement points. Therefore, while drafting a more precise and explicit set of guidelines, leaving room for expert judgement is also necessary. It is worth noting that clearer instructions do not necessarily equal stricter instructions, and with more mandatory sampling points it is more difficult to avoid the establishment of uncomfortable sites with possible limit value exceedances.*

*The simplification set in line 134 is meant to underline that the more there are measurement points the less important a single measurement point becomes. If an unlimited amount of measurement points was available, there would be no siting problem as it would be possible to cover the entire spatial domain with measurement points. Albeit not realistic, we believe it is something worth considering when taking into account the recent technological development.*

New Target Parameters

It is somewhat outside of my area, but I was surprised by the statement in line 144 that there is insufficient evidence on this point. Ultimately, though, I do not think that specifying pollen within the AAQD is aligned with its aims.

*The stated purpose of the AAQD is to protect human health and the environment as a whole. As reported by Durham et al., as many as one in four suffer from pollen-induced irritation symptoms in Europe each year. Moreover, the adverse health effects caused by pollen entail a substantial economic burden to societies in general (Zuberbier et al., 2014). It is against the self-proclaimed purpose of the AAQD to not state the need to measure pollen.*

*It is often argued that because pollen originates from plants (a natural source) there is nothing that can be done about it. It is true that the concentrations of pollen cannot be controlled, but multiple studies have shown how to model and forecast pollen concentrations (e.g. Muzalyova et al., 2021), and these methods can be used to reduce pollen exposure. Another frequent claim is that the AAQD is aimed specifically at anthropogenic emissions and therefore pollen falls outside of its scope. There are different constituents of air pollution that are being measured and which are of natural origin, for example SO2 from volcano eruptions and PM from sea salt and wildfires. This is acknowledged explicitly in the AAQD. Therefore, to exclude pollen just due to it having a natural source appears contradictory.*

*To clarify, the manuscript has been modified to underline that no limit values for pollen are being proposed.*

*Muzalyova, A., Brunner, J. O., Traidl-Hoffmann, C. and Damialis, A.: Forecasting Betula and Poaceae airborne pollen concentrations on a 3-hourly resolution in Augsburg, Germany: toward automatically generated, real-time predictions, Aerobiologia (Bologna)., 37(3), 425–446, doi:10.1007/s10453-021-09699-3, 2021.*

*Zuberbier, T., Lötvall, J., Simoens, S., Subramanian, S. V. and Church, M. K.: Economic burden of inadequate management of allergic diseases in the European Union: A GA2LEN review, Allergy Eur. J. Allergy Clin. Immunol., 69(10), 1275–1279, doi:10.1111/all.12470, 2014.*

As I have noted above, the AAQD is not prescriptive regarding how air quality is improved. Thus, not specifying that specific PM parameters are measured does not preclude this from happening. The absence of any reference to EMEP in this section is surprising. The position put seems to be that the AAQD is the most appropriate place to specify the need to measure additional PM parameters, but I do not feel that this case has been adequately made.

*As with the sensors, if the wider community already monitors additional parameters, we believe it is reasonable to propose this to be standardized through legislation*

*Whether the need and practical protocol outlining how to measure additional parameters is specified in the AAQD or in some other place (e.g. EMEP) is not a critical decision in our view. Factors favoring the AAQD include Commission's stated intent to align AAQD closer to that of the WHO guidelines, which now includes the BC and UFP parameters, and perhaps better overall visibility and accessibility.*

---

## Author Comment (AC2)

We thank the reviewers for taking time to read the article thoroughly and their valuable comments. We have answered all comments and we think the quality of the article has significantly improved in the review.

**Reviewer #2**

This article is a short opinion paper in which the authors propose three specific topics (use of sensors for regulatory monitoring, siting criteria and new air pollutants to monitor) that should be considered for the current revision of the European Air Quality directive. This article provides in my view a valuable input to this review process. It is well written and is also of interest to readers who are not directly involved in the review of the Air Quality Directive and even to readers who are not familiar with this legal framework document. This paper should therefore be published in ACP.

I have few comments that should be taken into account:

1. Page 2, last section: I don't really understand what the authors mean with the sentence. "Currently, the devices used for measurement-based online observation are almost exclusively limited to fixed measurement types." What are measurement-based online observations, and are the authors advocating for mobile measurements? Please check if this is clear or rephrase.

*'Measurement-based online observations' refers to observations that are made using actual measurement instruments (no modelling or objective estimation) and which produce data in real-time (concerns mainly PM monitoring; no manual filter replacement and weighing). The term 'fixed measurement' in this context refers to the most stringent and accurate measurement mode (see Table 1 in the preprint) and not the type (mobile or stationary) of the measurement. We are not advocating for mobile measurements to be included in the AAQD. The wording used in the AAQD for fixed (sometimes also referred to as 'continuous') and indicative measurement modes is slightly confusing, and perhaps not very well thought-out.*

*To be clearer, the last section of page 2 has been rephrased as:*

*"Currently, the devices used for measurement-based (fixed or indicative measurements; no modelling or objective estimation) online observations are almost exclusively limited to the most accurate fixed measurement types."*

Then the authors state that type approval is mandatory for regulatory measurements. I think this is not correct (but maybe I'm wrong): Regulatory measurements must be done according to the reference method as defined in EN standards, and the easiest way to do this is by using an instrument that is type approved according to the corresponding EN standard. However, any other method can be used which give results that are equivalent to the reference method. Of course, equivalence needs to be demonstrated (see the Air Quality Directive). I therefore also don't think that the conclusion is correct that there is no "incentive for a company … (page 3, first line)".

*This is an unclear point in the AAQD, and we are not entirely sure what is the correct interpretation. As noted by the reviewer, it is said in the AAQD that a reference method OR a method equivalent to the reference method can be used if the equivalence is demonstrated to be sufficient (not necessarily type-approved). We have discussed this issue with the Finnish National Reference Laboratory (NRL) for Air Quality, and our understanding is that while type-approval may not be an absolute necessity it is, however, a standard practice to obtain one from the TUV. We speculate there are a few reasons for*

*this. 1) Once obtained, the type-approval is valid in all Member States. This means that the instrument can be used in other EU countries with the same type-approved certification status. The instrument must still undergo a demonstration of equivalence, which means that the suitability of the instrument for the specific measurement environment, where it is about to be used, is ensured. To our understanding, this demonstration is different and less laborious than the one alternative to the type-approval process. 2) According to the Guide to demonstration of equivalence of ambient air monitoring methods (https://ec.europa.eu/environment/air/quality/legislation/pdf/equivalence.pdf), the demonstration of equivalence should be carried out by a laboratory nominated by the Member State's National Competent Authority (NCA), which typically means that the NRL conducts the testing. When coupled with the notion that the process of demonstrating equivalence appears to be similar to that of the corresponding EN reference standards, it is likely that the obtaining of a type-approval and the equivalence demonstration without type-approval are equally heavy processes. Therefore, it may be more appealing for the companies to proceed with the formal type-approval route.*

*This issue pointed by the reviewer is valid, and the following change was made:*

*"This is at least partly due to the long and costly process of acquiring a device type- approval, which is  common practice if regulatory measurements are to be made."*

2. The authors suggest that low-cost sensors can/should play a role in the hierarchical network of regulatory observations, and they somehow imply that sensors are capable for providing indicative measurements (Figure 1). I think, however, that this is what we currently expect or maybe hope, but to my knowledge the current literature does not show that sensors are in real–world applications and over longer time periods capable of providing data of higher quality than modelling or objective estimation. Anyhow, it is clear that a good data quality with sensors will not come for free, appropriate strategies for quality assurance and control must be implemented that will lead to significant operational costs. Maybe the authors want to mention this.

*It is fair to say that low-cost sensors have their problems, and instead of gaining better understanding of the measured air quality, the use of low-cost sensors may in some cases lead to even more confusion and uncertainty due to the lack information regarding what the sensors are truly measuring. We slightly disagree on the statement that the literature shows no examples of successful sensor deployments where objective estimation, and to some degree modelling, would be a more appropriate way of assessing air quality. There are certainly weaknesses associated with these observation modes as well. It is true that improved quality will likely lead to increased overall costs as well. However, the key point and an open-ended question is whether the cost-benefit ratio of sensors would still indicate them to be a justifiable addition to the list commonly used measurement devices. It is worth noting that one of the main factors driving low-cost sensors forward has been the high unit and operational costs of the conventionally used instruments.*

*Following comment was added to the manuscript page 3:*

*"It is likely that the cost of owning and operating sensors would increase as a result of formal performance testing and improved overall quality. However, we believe that standardization is the most appropriate way to proceed at this stage, and it remains to be seen how the cost structure of sensors eventually compares to the conventional high-cost instruments."*

3. In the summary and conclusions section it is stated that "technological development of air quality sensors is advancing rapidly". True, there is significant technological development in PM sensors, or the miniaturization of optical particle counting technologies. However, for gas sensing the applied technologies in air quality sensors (e.g. electrochemical cells and metal-oxide sensors) are not new and improvements are rather slow. Maybe we are misled by the fast technological developments in sensor integration, data communication, storage, and visualization etc. Given the improving air quality in Europe (e.g. NO2), technological developments of sensors are needed so that they can play a role in regulatory air quality monitoring. If the authors agree, they could add a sentence.

*We agree. The sentence has been rephrased as:*

*"Technological development of air quality sensors is advancing rapidly but more is needed for the sensors to have a major role in regulatory air quality monitoring."*

4. Page 3, 2nd paragraph. I don't the first sentence "Although the testing protocol …", is linguistically correct, please check.

*The sentence was rephrased as follows:*

*"Although the testing protocol should be less exhaustive in order for companies to apply for device type-approval at a lower threshold, it is equally important that no major compromises are being made with respect to the quality criteria of the testing protocol; …"*

5. Page 4, line106, typo, should be "zones".

*Corrected.*

---

## Author Response (AR2)

Dear Editor,

we wish to thank the reviewers for taking their time to read and comment our article. We corrected the minor point presented by reviewer #2 but we did not respond to the major comment. We understand the drawbacks of the proposed solution, but in our view there is no clear way in which compromises could be avoided altogether. We have decided that it is better to propose something rather than nothing.

On behalf of all authors,
Joel Kuula
Atmospheric Composition Research
Finnish Meteorological Institute
joel.kuula@fmi.fi